# Sphingosine Kinase 1 Deficiency in Smooth Muscle Cells Protects against Hypoxia-Mediated Pulmonary Hypertension via YAP1 Signaling

**DOI:** 10.3390/ijms232314516

**Published:** 2022-11-22

**Authors:** Jiwang Chen, Angelia Lockett, Shuangping Zhao, Long Shuang Huang, Yifan Wang, Weiwen Wu, Ming Tang, Shahzaib Haider, Daniela Velez Rendon, Raheel Khan, Bing Liu, Nicholas Felesena, Justin R. Sysol, Daniela Valdez-Jasso, Haiyang Tang, Yang Bai, Viswanathan Natarajan, Roberto F. Machado

**Affiliations:** 1Department of Medicine, Section of Pulmonary, Critical Care Medicine, Sleep and Allergy, University of Illinois at Chicago, Chicago, IL 60612, USA; 2Center for Cardiovascular Research, University of Illinois at Chicago, Chicago, IL 60612, USA; 3Division of Pulmonary, Critical Care, Sleep, and Occupational Medicine, Indiana University School of Medicine, Indianapolis, IN 46202, USA; 4Department of Pharmacology and Regenerative Medicine, University of Illinois at Chicago, Chicago, IL 60612, USA; 5Department of Bioengineering, University of Illinois at Chicago, Chicago, IL 60612, USA; 6Department of Medicine, Jacobs School of Medicine and Biomedical Sciences, The University at Buffalo, Buffalo, NY 14260, USA; 7Department of Bioengineering, University of California San Diego, La Jolla, CA 92093, USA; 8State Key Laboratory of Respiratory Disease, Guangzhou Institute of Respiratory Disease, First Affiliated Hospital of Guangzhou Medical University, Guangzhou 510030, China

**Keywords:** sphingosine kinase 1, S1P, pulmonary artery smooth muscle cell, proliferation, YAP1, verteporfin, pulmonary hypertension

## Abstract

Sphingosine kinase 1 (SPHK1) and the sphingosine-1-phosphate (S1P) signaling pathway have been shown to play a role in pulmonary arterial hypertension (PAH). S1P is an important stimulus for pulmonary artery smooth muscle cell (PASMC) proliferation and pulmonary vascular remodeling. We aimed to examine the specific roles of SPHK1 in PASMCs during pulmonary hypertension (PH) progression. We generated smooth muscle cell-specific, *Sphk1*-deficient (*Sphk1^f/f^ TaglnCre^+^*) mice and isolated *Sphk1*-deficient PASMCs from *SPHK1* knockout mice. We demonstrated that *Sphk1^f/f^ TaglnCre^+^* mice are protected from hypoxia or hypoxia/Sugen-mediated PH, and pulmonary vascular remodeling and that *Sphk1*-deficient PASMCs are less proliferative compared with ones isolated from wild-type (WT) siblings. S1P or hypoxia activated yes-associated protein 1 (YAP1) signaling by enhancing its translocation to the nucleus, which was dependent on SPHK1 enzymatic activity. Further, verteporfin, a pharmacologic YAP1 inhibitor, attenuated the S1P-mediated proliferation of hPASMCs, hypoxia-mediated PH, and pulmonary vascular remodeling in mice and hypoxia/Sugen-mediated severe PH in rats. Smooth muscle cell-specific SPHK1 plays an essential role in PH via YAP1 signaling, and YAP1 inhibition may have therapeutic potential in treating PH.

## 1. Introduction

Pulmonary arterial hypertension (PAH) is a severe and progressive disease that results in increased pulmonary vascular resistance (PVR), right heart failure, and, ultimately, death. Sustained pulmonary vasoconstriction, excessive pulmonary vascular remodeling, and thrombosis, in situ, are the three major causes of elevated PVR in patients with PAH [1,2,3]. Pulmonary vascular remodeling is characterized, in part, by significant medial and intimal hypertrophy due to the enhanced proliferation of pulmonary artery smooth muscle cells (PASMCs) and their resistance to apoptosis. Most of the current therapies in PAH target pulmonary vasoconstriction, and there is a need to develop new targets to combat pulmonary vascular remodeling [2]. 

We previously showed that sphingosine kinase 1 (SPHK1) is a potential novel target for PAH therapy [4]. SPHK1 catalyzes the phosphorylation of sphingosine to generate sphingosine-1-phosphate (S1P), a potent bioactive sphingolipid mediator and a known regulator of cell proliferation, differentiation, motility, Ca^2+^ mobilization, and endothelial barrier regulation [5,6,7]. Our previous studies showed that SPHK1 was upregulated in PASMCs isolated from patients with PAH and that the genetic deletion of *Sphk1* in mice or the pharmacological inhibition of SPHK1 activity reduced hypoxia-induced pulmonary hypertension (PH) in mice and rats [4]. An earlier study showed that serum S1P induces yes-associated protein (YAP1) nuclear localization through S1P receptor 2 (S1PR_2_) and modulates the proliferation of HEK293A cells [8]. In airway smooth muscle cells, S1P stimulates cell proliferation, migration, and contraction by binding to S1PR_2/3_ and signaling through the ROCK/YAP/FOXM1 axis [9]. It is unknown, however, whether SPHK1 in PASMCs is upregulated in experimental PH and whether PASMC SPHK1 is specifically involved in the development of PH.

To address these questions, we examined the role of PASMC-specific *Sphk1* in vivo and in vitro, as well as the potential role of SPHK1-dependent YAP1 signaling in the development of pulmonary vascular remodeling and PAH.

## 2. Results

### 2.1. SPHK1 Is Upregulated in the PASMCs of Rodent Models of PH

Lung tissue immunofluorescence staining demonstrated that the protein levels of SPHK1 were significantly upregulated in the pulmonary artery medial layer smooth muscle cells from different rodent models of pulmonary hypertension, including chronic hypoxia-mediated PH in mice, monocrotaline (MCT)-mediated PH, and hypoxia-mediated PH (HPH) in rats (Figure 1A–D). 

### 2.2. Smooth Muscle Cell-Specific Sphk1 Knockout Mice Are Protected from Hypoxia-Induced PH 

Having demonstrated elevated SPHK1 expression in the lung tissues of rodent PH models, next, we investigated the effect of *Sphk1* deletion in SMCs on the development of experimental PH in mice. The smooth muscle cell-specific knockdown of *Sphk1* in PASMCs was achieved by breeding *Sphk1^f/l^* mice with Tagln Cre^+^ mice in the C57BL6 background. The F2 generation mice were genotyped and their lungs were immunostained to confirm the deletion of SPHK1 expression in PASMCs. *Sphk1* mRNA expression was measured in PASMCs isolated from the lung. Compared with control mice (*Sphk1^f/f^*), *Sphk1^f/f^ TaglnCre^+^* mice were constitutively deficient in SPHK1 in the smooth muscle cell layer (Appendix A). The deletion of *Sphk1* in smooth muscle did not induce PH under normoxic conditions (Figure 2). However, *Sphk1^f/f^ TaglnCre^-^* exposed to hypoxia had increased markers of HPH compared with normoxia-exposed mice as right ventricular systolic pressure (RVSP); right ventricular hypertrophy (RVH, assessed by RV/(LV + S) ratio); and pulmonary vascular remodeling were increased. HPH development was inhibited in the *Sphk1^f/f^ TaglnCre^+^* mice exposed to hypoxia for four weeks, as they exhibited lower RVSP, decreased RVP, decreased RVH, and decreased pulmonary vascular remodeling compared with their hypoxia-exposed wild-type siblings, *Sphk1^f/f^ TaglnCre^−^* mice (Figure 2). 

### 2.3. Smooth Muscle Cell-Specific Sphk1 Knockout Mice Are Protected from Hypoxia Plus Sugen-Mediated Pulmonary Hypertension

To further examine whether *Sphk1^f/f^TaglnCre^+^* mice are protected from PH, a mouse model of hypoxia/Sugen (an inhibitor of VEGFR)-mediated PH was also used. The *Sphk1^f/f^ TaglnCre^+^* mice and their WT siblings were exposed to hypoxia for four weeks. Four doses of Sugen (20 mg/kg body weight) were administered subcutaneously to the mice once per week. Under normoxia/Sugen conditions, there was no *Sphk1* deletion effect on PH development. Hypoxia/Sugen led to PH development in *Sphk1^f/f^TaglnCre^−^* mice compared with normoxia/Sugen control mice. *Sphk1^f/f^TaglnCre^+^* mice exposed to hypoxia/Sugen exhibited lower RVSP, decreased RVH (assessed by the RV/(LV + S) ratio), and less severe pulmonary vascular remodeling compared with hypoxia-exposed *Sphk1^f/f^ TaglnCre^-^* mice (Figure 3). Taken together with Figure 2, these results suggest that the hypoxia-induced expression of SPHK1 in PASMCs plays an essential role in the development of PH in vivo.

### 2.4. PASMCs Isolated from Sphk1^-/-^ Mice Are Less Proliferative and Less Viable

To investigate whether *Sphk1* deficiency plays a role in cell proliferation, viability, and cell death, we isolated PASMCs from *Sphk1^-/-^* global knockout, and age-matched WT mice. As shown in Figure 4, at baseline, PASMCs from *Sphk1^-/-^* mice compared with WT mice were less proliferative, less viable, and had increased cell death. Hypoxia increased proliferation and cell viability in WT PASMCs compared with normoxia controls. Furthermore, PASMCs isolated from *Sphk1^-/-^* mice and exposed to hypoxia exhibited lower viability and higher cell death compared with cells isolated from hypoxia-exposed WT cells. These results show that SPHK1 regulates cell proliferation, viability, and apoptosis in PASMCs.

### 2.5. Hypoxia Induces YAP1 and TAZ Expression in the Lungs of Rodent Models of PH

Recent studies suggest a role for HIPPO/yes-associated protein (YAP1) signaling in the pathogenesis of PAH [10,11,12,13]; however, how hypoxia stimulates HIPPO/YAP1 signaling in PASMCs is unclear. To determine the link between hypoxia and YAP1 expression, lung tissues from the controls and three rodent models of PH were investigated. As shown in Figure 5A, lung tissues from mouse HPH showed a trend in increased YAP1 expression; however, rat MCT and rat Sugen + hypoxia showed significantly higher levels of YAP1 expression compared with control lungs (Figure 5B,C). In addition to YAP1, hypoxia and hypoxia plus Sugen enhanced the expression of transcriptional coactivator with PDZ motif (TAZ) (Figure 5D), which functions as a coactivator with YAP1 in the HIPPO pathway. These results suggest a plausible link between SPHK1 and YAP1 in mouse HPH.

### 2.6. Hypoxia and S1P Stimulate YAP1 Nuclear Translocation in PASMCs

In most nonconfluent mammalian cells, YAP1 is predominantly localized in the cytosol. However, the dephosphorylation of YAP1 at Ser 127 and the activation of cells result in the nuclear translocation of YAP1 [14]. As hypoxia stimulates a plethora of transcriptional factors and signal transduction pathways, next, we investigated the effect of hypoxia on YAP1 localization. Exposing hPASMCs to hypoxia (30 min) significantly increased YAP1 translocation to the nucleus, as determined by immunofluorescence (Figure 6A,B). Similarly, the S1P treatment of PASMCs induced YAP1 nuclear translocation, an effect that was abrogated by concomitant treatment with the YAP1 activation inhibitor verteporfin (Figure 6C,D). S1P also regulated YAP1 expression via the activation of the S1PR2 pathway as the inhibition of S1PR2 with JTE-013 attenuated the S1P-mediated increase in YAP1 expression (Figure 6E,F). In contrast to YAP1, the S1P-mediated stimulation of TAZ expression in hPASMCs was only marginal and not statistically significant (Figure 6E,G). To test the dependency of SPHK1 enzymatic activity on hypoxia-mediated YAP1 activation, primary hPASMCs were transfected with an adenovirus control vector (AdCon) or dominant negative SPHK1 (AdSPHK1DN), which lacks kinase activity [15]. In the presence of AdCon, hypoxia promoted YAP1 nuclear translocation (Figure 6H,I). However, under both normoxic and hypoxic conditions, YAP1 nuclear translocation was attenuated in the presence of AdSPHK1DN, suggesting that the kinase activity of SPHK1 is necessary for YAP1 translocation. To further assess the role of SPHK1 in YAP1 regulation, HPASMCs were treated with control or *SPHK1* siRNA for 72 h. As shown in Figure 7A–C, silencing *SPHK1* expression via siRNA decreased both the total cellular YAP1 expression and its nuclear translocation. SPHK1 knockdown was confirmed by assessing both mRNA and protein expression (Figure 7D–F). 

### 2.7. S1P Promotes YAP1 Nuclear Translocation via Ligation to S1PR2

To test whether S1P promotes YAP1 nuclear translocation via the ligation of S1PR2, primary hPASMCs were treated with control or S1PR2 siRNA for 72 h and then stimulated with S1P. As shown in Figure 8A–C, in the presence of control siRNA, S1P increased both YAP1 expression and nuclear localization. However, silencing S1PR2 expression via siRNA decreased both basal and S1P-mediated YAP1 expression and nuclear translocation. The effectiveness of S1PR2 knockdown was assessed by measuring both its mRNA and protein expression (Figure 8D–F). To confirm the role of S1PR2 in the S1P-mediated regulation of YAP1 activation and localization, S1PR2 was inhibited with JTE-013. Pretreatment with JTE-013 significantly reduced S1P-induced YAP1 nuclear translocation (Figure 8G,H). Together, these data indicate that S1P promotes YAP1 nuclear localization in HPASMCs via the ligation of S1PR2.

### 2.8. YAP1 Regulates S1P-Mediated Proliferation in hPASMCs

The above findings point to a role for SPHK1/S1P/S1PR2 signaling in hypoxia-induced YAP1 expression and translocation to the cell nucleus, and our earlier observation indicates that the SPHK1/S1P pathway plays a key role in PASMC proliferation [4]. Therefore, we next assessed whether silencing or inhibiting YAP1 regulates apoptosis and cell proliferation. As shown in Figure 9A,B, stimulating cells with S1P had no effect on apoptosis, as assessed by TUNEL assays, while inhibiting YAP1 with verteporfin (200 nM) induced hPASMC apoptosis. Given our observation that SPHK1 deficiency also increases cell death (Figure 4), we assessed the effect of S1P on apoptosis when YAP1 activity is inhibited and found that S1P treatment did not reverse the effects of verteporfin. Furthermore, downregulation of YAP1 expression by siRNA or treatment with verteporfin attenuated S1P-mediated cell proliferation (Figure 9C,D). Taken together, these results suggest that YAP1 represses apoptosis and promotes pro-survival phenotypes induced by SPHK1/S1P signaling in hPASMCs.

### 2.9. YAP1 Inhibition with Verteporfin Prevents Hypoxia-Mediated PH and Pulmonary Vascular Remodeling in Mice

Having demonstrated that YAP1 is upregulated in experimental PH and inhibition of YAP1 attenuates proliferation and promotes PASMC apoptosis, we reasoned that the inhibition of YAP1 might attenuate the development of hypoxia-mediated PH in mice. When compared with vehicle treatment, verteporfin treatment (4.5 mg/kg body weight every 48 h for 4 weeks) attenuated hypoxia-induced increases in RVSP, right ventricular hypertrophy (RV/(LV + S)), and pulmonary vascular remodeling (Figure 10) without significant changes in the basal levels of these parameters in normoxic conditions. 

### 2.10. YAP1 Inhibition with Verteporfin Attenuates Hypoxia/Sugen-Mediated PH and Pulmonary Vascular Remodeling in Rats

The rat model of PH mediated by Sugen and hypoxia induces severe pulmonary vascular remodeling characterized by obliterative pulmonary vascular lesions with neointima formation in distal pulmonary arteries. Rats were administered Sugen (20 mg/kg subcutaneously once), followed by exposure to chronic hypoxia (10%, 3 weeks, n = 6 pre-treatment group). During the hypoxia exposure, verteporfin (4.5 mg/kg for 3 weeks) or vehicle was administered every 48 h. After the 3-week treatment, the mice were kept in normoxia for another 2 weeks. Compared with the controls, the verteporfin treatment attenuated the development of PH, as assessed by changes in RVSP, RVH, and the development of medial hypertrophy and neointima formation (Figure 11A–E). The verteporfin treatment also decreased cell proliferation and obliterative lesions (Figure 11F). Together with our observations in the HPH mouse model, these results suggest that YAP1 may be a novel candidate to be used as a target for therapeutic intervention in PAH.

## 3. Discussion

In this study, we show that SPHK1 is upregulated in the medial layer of pulmonary arteries in chronic hypoxia or hypoxia + Sugen-mediated PH in mice and rats and in MCT-mediated PH in rats and that smooth muscle cell *Sphk1* knockout mice are protected from hypoxia or hypoxia + Sugen-mediated PH. In addition, we report that the SPHK1/S1P/S1PR2 axis activates YAP1 signaling and that the pro-survival effects of this axis in PASMCs are dependent on YAP1. Finally, YAP1 inhibition attenuates experimental PH. 

We previously established the role of the SPHK1/S1P/S1PR2 signaling axis in PAH [4,16,17]. Hypoxia and other stimuli, such as growth factors, upregulate SPHK1 expression, increasing S1P production in PASMCs [4]. S1P ligates S1PR2 on the PASMC surface, leading to the phosphorylation of both ERK and the signal transducer and activator of transcription (STAT) 3, both known to induce cell proliferation [18]. Our current data add to these observations and suggest that, both under normoxia and hypoxia, S1P also promotes PASMC pro-survival phenotypes in a YAP1-dependent manner via the ligation of S1PR2. 

Other studies have also established a link between S1P/S1PR2 and YAP1 signaling [8,9]. In vivo, both hypoxia and hypoxia + Sugen enhanced YAP1 expression in PASMCs, which was SPHK1 dependent. The role of Sugen in enhancing hypoxia-induced YAP1 is unclear, but most likely, it is due to higher SPHK1 expression and increased S1P production. It is also unclear if Sugen alone has any effect on SPHK1 expression in mouse PASMCs.

Gairhe et al. [19] showed increased SPHK1 and SPHK2 expression as well as S1P in the remodeled pulmonary arteries of patients with idiopathic PAH and the Sugen/hypoxia model of PAH in rats. Further, the inhibition of SPHK1 reduced S1P levels and occlusive pulmonary arteriopathy without reducing RVSP or right ventricular hypertrophy. This is in contrast to our current finding that the specific knockdown *of Sphk1* in PASMCs reduces hypoxia and hypoxia + Sugen-induced RVSP. Our data also suggest the involvement of S1PR2 in hypoxia and S1P-mediated PASMC proliferation. It will be interesting to determine if Sugen alone or hypoxia + Sugen has any effect on the expression of S1PR2 and other S1PRs in PASMCs in the context of PAH development. In hepatoma cells, the S1P/S1PR2/YAP1 signaling axis stimulates cell proliferation by upregulating connective tissue growth factor [20]. We have also shown a role for SPHK1/S1P signaling in TGF-β-induced YAP1 activation and mitochondrial ROS generation, resulting in fibroblast activation, a critical driver of pulmonary fibrosis [11]. Several lines of evidence support a role for YAP1 in PH and pulmonary vascular remodeling [10,12]. Our data indicate a modest but statistically insignificant increase in TAZ expression in response to the S1P treatment of PASMCs, whereas hypoxia led to an increase in both YAP1 and TAZ expression in the lung and in PASMCs. These observations suggest that S1P may alter YAP1/TAZ interactions without altering TAZ expression or that the hypoxia-mediated upregulation of TAZ is independent of the S1P signaling axis. Future studies using SPHK1 conditional knockout mice are necessary to investigate these differences. However, other studies have linked YAP1/TAZ signaling to PAH via mechanotransduction and the vascular metabolic reprogramming that promotes pulmonary vascular remodeling [10,12]. Kudryashova et al. [13] showed that large tumor suppressor 1 (LATS1) inactivation and the consequent YAP1 upregulation were required for mTOR downstream signaling, as well as increased proliferation and the survival of human PAH PASMCs. Our in vitro studies also confirmed observations suggesting a role for YAP1 and STAT3 signaling in the proliferative effects of S1P in PASMCs [21,22,23]. 

Our results support the growing body of evidence suggesting YAP1 is a potential therapeutic target in PAH. YAP1 expression is increased in PAH PASMCs, and experimental PH and the knockdown of YAP1 and TAZ markedly induce COX-2 expression and downstream prostaglandin production [12]. We show here that hypoxia stimulated the expression of YAP1 and TAZ while S1P enhanced YAP1 but not TAZ expression in hPASMCs. Bertero et al. [11] showed that verteporfin treatment at the time of PH induction decreased YAP1 expression and downstream signaling and prevented the development of MCT-induced PH in rats. The inhibition of the E3 ubiquitin ligase seven-in-absentia-homolog 2 (Siah2), inhibits LATS 1/2 degradation and YAP1 activation and prevents the development of MCT-induced PH in rats [24]. Our data add strength to these observations since we show that YAP1 inhibition attenuates the progression of established diseases in a different experiential model of severe pulmonary vascular remodeling and PH. 

In summary, we have demonstrated that the SPHK1/S1P/YAP1 signaling axis in PASMCs is critical to the development of pulmonary vascular remodeling. More importantly, our results support the growing body of evidence that SPHK1, S1P, and YAP1 are potential therapeutic targets for PAH.

## 4. Materials and Methods 

### 4.1. Reagents, Drugs, and Antibodies

Cy3^TM^-labeled mouse anti-α-smooth muscle actin antibody (C6198) and HRP-labeled monoclonal anti-β-actin (A3854) antibody were obtained from Sigma (St Louis, MO, USA). Alexa Fluor^®^ 488 Donkey anti-Rabbit IgG antibody (R37118) was obtained from Life Technologies Inc. Rabbit anti-SPHK1 antibody (ab71700), and mouse monoclonal anti-TAZ antibody (ab242313) were purchased from abcam^®^ (Cambridge, MA, USA). Mouse anti-human S1PR2 antibody (sc-365963) was purchased from Santa Cruz Biotechnology, Inc. (Dallas, TX, USA). Horseradish peroxidase (HRP)-linked anti-rabbit IgG antibodies were obtained from Santa Cruz Biotechnology, Inc. (Santa Cruz, CA, USA). Rabbit anti-YAP1 antibody (14074) was purchased from Cell Signaling Technology (Danvers, MA, USA). 

### 4.2. Mouse PASMC Isolation

PASMCs were isolated from mouse lungs using a modified method created by Marshall et al. [25]. Solution I (a mixture of 5 mL of medium 199 (M199) growth medium containing 0.025 g of low-melting-point agarose type VII (Sigma, St. Louis, MO, USA) and 0.025 g of iron beads (diameter < 10μM; Sigma, St. Louis, MO, USA)) was slowly injected through the RV, thereby perfusing the PA. Solution II (M199 growth medium (1 mL) containing 0.025 g of agarose type VII) was injected into airways through the trachea. The lungs were then immersed in cold PBS to cause the agarose gel to cool. All the lobes were then isolated and finely minced in a Petri dish. The suspension was then mixed in M199 growth medium containing 80 U/mL of type IV collagenase (Sigma, St. Louis, MO, USA) and incubated at 37 °C for 90 min in a 100 mm culture plate. The tissue was further disrupted by being passed through a 16-gauge followed by an 18-gauge needle approximately three times. With the use of a magnetic column (Invitrogen, Carlsbad, CA, USA), the arteries or arterial tissues containing the iron beads were collected. The supernatant was aspirated, and the arteries were washed and suspended in 5 mL of M199 containing 20% FBS. Aliquots of the suspension were transferred to T25 culture flasks and the medium was changed using a magnetic column twice a week for up to two and a half weeks. Smooth muscle cell purity was determined with immunostaining using a cy3-labeled smooth muscle-specific α-actin antibody (Sigma, St. Louis, MO, USA).

### 4.3. Cell Proliferation, Viability, Apoptosis, and Lactate Dehydrogenase Assays 

Cell proliferation was determined using either a 5-bromo-2′-deoxyuridine (BrdU) incorporation assay or cell counting. BrdU assays (QIA58, Calbiochem, San Diego, CA, USA) were performed in a 96-well format according to manufacturer’s instructions using starting cell densities of 4000 cells/well. For cell counting, cells were seeded in a 6-well culture plate (40,000 cell/well) and cultured in a normoxia cell culture incubator. After 72 h, the cells were trypsinized and counted with a TC10^TM^ automated cell counter (Bio-Rad, Hercules, CA, USA). Cell viability was determined using the Cell Titer 96 AQueous One Solution Cell Proliferation Assay (Promega Corporation, Madison, WI, USA). Cell death was measured with the Cytotoxicity Detection Kit (using lactate dehydrogenase; LDH) (Roche, Mannheim, Germany). Cell apoptosis was examined using an in situ BrdU–Red DNA fragmentation (TUNEL) kit. The cells on the coverslips were examined using a Nikon Eclipse E800 fluorescence microscope, and the images were processed by the MetaMorph software (Molecular Devices, Inc., San Jose, CA, USA). Approximately ten images were taken from each condition, and over 500 cells were counted according to DAPI staining.

### 4.4. Lung Tissue Immunofluorescence Staining

Paraffin-embedded lung tissue sections were deparaffinized with xylene and rehydrated. Antigen retrieval was used before blocking in PBS with 10% normal goat serum, 0.1%BSA, 0.3% TX-100. The antigen retrieval solution was Tris-EDTA buffer (10 mM Tris Base; 1 mM EDTA solution; 0.05% Tween 20; pH 9.0). The dilutions for rabbit against SPHK1 (Abcam^®^, catalog# ab71700) and cy3-labeled mice against smooth muscle actin are 1:100 and 1:300, respectively. The secondary antibody for SPHK1 was Alexa Fluor^®^ 488 Donkey anti-Rabbit IgG antibody (1:500). An anti-fade mounting media with DAPI (Life Science Inc., St. Petersburg, FL, USA) was used to fix the coverslip to a slide. The slides were examined using a Nikon Eclipse E800 fluorescence microscope, and the images were processed by the MetaMorph software (Molecular Devices, Inc., San Jose, CA, USA).

### 4.5. Adenoviral Constructs of Wild-Type and Dominant-Negative SPHK1 Plasmids

The *SPHK1* WT and dominant–negative mutant were generated as reported previously [4]. All constructs were confirmed with DNA sequencing.

### 4.6. Transfection of Plasmid DNA and Small Interfering RNA in hPASMCs 

Human PASMCs (hPASMCs) transfected with scrambled small interfering RNA (siRNA) (D-001810-02), Human *SPHK1* siRNA-SMARTpool (M004172-03), human *S1PR2* siRNA-SMARTpool (L-003952-00), and *YAP1* siRNA (M-012200-00) were purchased from Dharmacon (Thermo Fisher Scientific, Lafayette, CO, USA) and transfected into primary hPASMCs using DharmaFECT 1 transfection reagent (Thermo Fisher Scientific, Waltham, MA, USA) according to the manufacturer’s protocol. 

### 4.7. Animal Models of Pulmonary Hypertension and Hemodynamic Measurements

All experiments were approved by the Ethics and Animal Care Committee of the University of Illinois at Chicago and Indiana University. In the rodent models of hypoxia-mediated pulmonary hypertension, eight-week-old male *Sphk1^f/f^ TaglnCre^+^* mice in the C57BL/6 background and their WT siblings were exposed to hypoxia (10% O_2_) in a ventilated chamber for four weeks. In a rat model of hypoxia plus Sugen-mediated severe PH, male Sprague–Dawley rats (190–200 g) from Charles River were used as previously described [1,26]. One dose of Sugen (20 mg/kg body weight) was injected into the rats subcutaneously immediately before a three-week hypoxia exposure (10% O_2_). After hypoxia exposure, the rats were placed back in normoxia for another three weeks, and verteporfin (4.5 mg/kg body weight, IP) or vehicle was injected once every other day for three weeks.

At the end of the experiments, RVSP was determined with right heart catheterization using a Millar pressure transducer catheter. A weight ratio of the right ventricle divided by the sum of the left ventricle and septum (RV/(LV + S)) was measured to determine the extent of RVH. Animals were anesthetized with ketamine/xylazine and placed on a heating pad. A midline incision was made on the neck, and a portion of the right external jugular vein was carefully exposed. The distal portion of the jugular vein was ligated using a suture thread, and a small incision was made in the jugular vein. A Millar Mikro-tip catheter–transducer (model PVR-1030 for mice; model SPR-869NR for rats) was inserted into the right ventricle (RV) via the right jugular vein. Data were continuously recorded by the MPVS-300 system with a Powerlab A/D converter (AD Instruments, Colorado Springs, Colorado. RV systolic pressure and heart rate were measured and calculated. After recording, the animals were sacrificed via exsanguination, and then lung tissue was harvested for histological studies.

Pulmonary artery remodeling was assessed using Aperio ImageScope software (version 11) after lungs were stained with hematoxylin and eosin. A minimum of 10 microscopic fields were examined for each slide. Approximately twenty muscular arteries with diameters (D) of 50–100 μm or D < 50 μm per lung section were outlined and measured. Vessel remodeling was calculated as ((external vessel area-internal vessel area) / external vessel area), as previously described [27]. 

### 4.8. Statistical Analysis

Statistical analysis of experimental data was performed using GraphPad Prism 5.1 (GraphPad Software, Inc., La Jolla, CA, USA). Results are expressed as mean ± SEM from at least three experiments. Student’s *t*-test and analysis of variance were used to compare two and three groups, respectively. Dunnett’s test was utilized for multiple comparison corrections and adjustments. An adjusted *p* less than 0.05 was considered statistically significant.

## Figures and Tables

**Figure 1 ijms-23-14516-f001:**
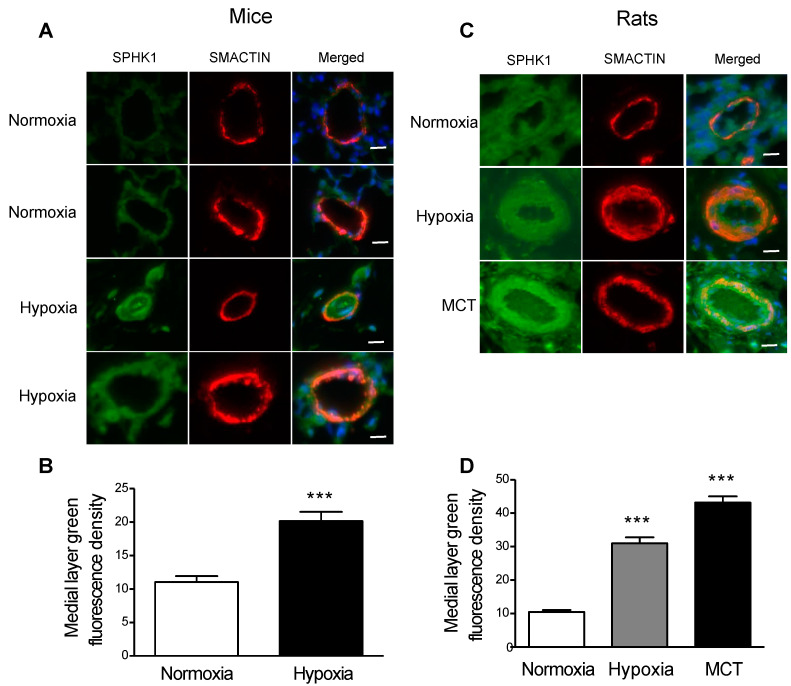
Lung immunofluorescence staining demonstrates increased SPHK1 expression levels in pulmonary artery smooth muscle cells in rodent models of pulmonary hypertension. (**A**) Mouse model of hypoxia-mediated pulmonary hypertension; (**B**) quantification data of medial layer SPHK1 expression levels for (**A**) via Image J version 1.4; (**C**) rat models of pulmonary hypertension mediated by monocrotaline or hypoxia; (**D**) quantification data of medial layer SPHK1 expression levels for (**C**) via Image J. *n* = 3; scalebar: 20 µm. *** *p* < 0.001 versus normoxia group.

**Figure 2 ijms-23-14516-f002:**
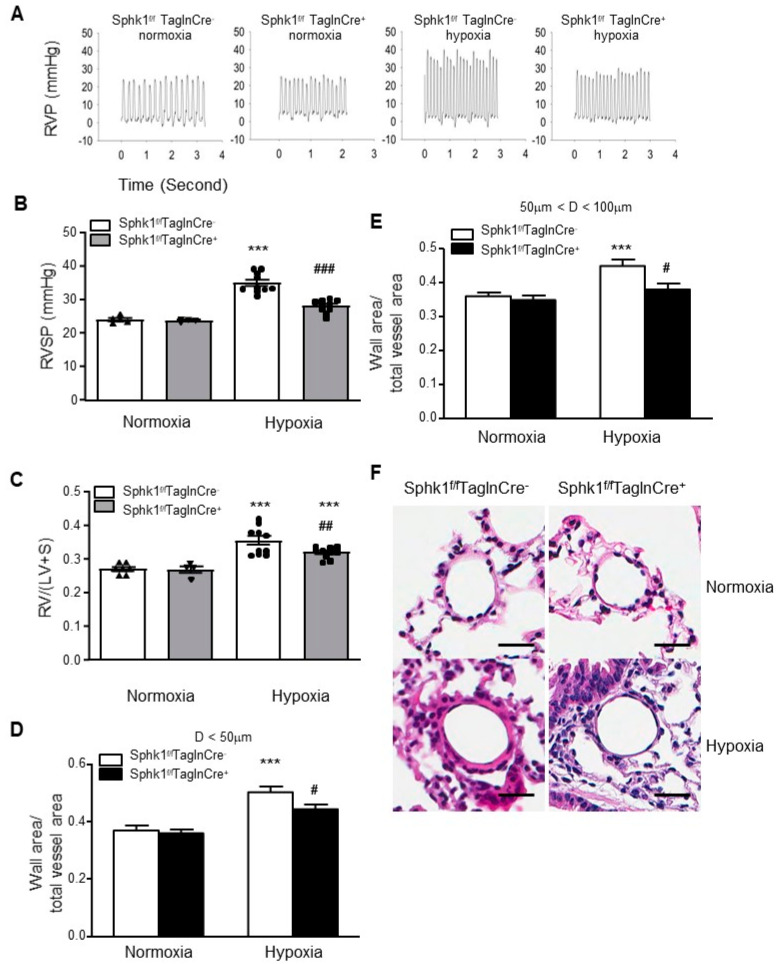
Smooth muscle cell-specific *Sphk1* knockout (*Sphk1^f/f^TaglnCre^+^*) mice are protected from hypoxia-induced pulmonary hypertension. When compared with their wild-type siblings (*Sphk1^f/f^TaglnCre^−^*), *Sphk1^f/f^TaglnCre^+^* mice exposed to hypoxia developed (**A**,**B**) less severe elevations in right ventricular systolic pressure (RVSP), (**C**) less right ventricular hypertrophy (RV/(LV + S)), and (**D**) lower increases in the ratios of wall area to the total vessel area of pulmonary arteries less than 50 μm (**E**) and 50–100 μm in diameter. (**F**) Representative pulmonary artery images in the lung sections of control and *Sphk1^f/f^TaglnCre^+^* mice exposed to normoxia or hypoxia. Scalebar: 20 μm. Results are expressed as mean ± SEM; number of mice in each group (*n* = 5–10) is shown in (**B**,**C**). *** *p* < 0.001 versus normoxia *Sphk1^f/f^TaglnCre-* group; #, ##, ### *p* < 0.05, hypoxia *Sphk1^f/f^TaglnCre^−^* vs. hypoxia *Sphk1^f/f^TaglnCre^+^* groups.

**Figure 3 ijms-23-14516-f003:**
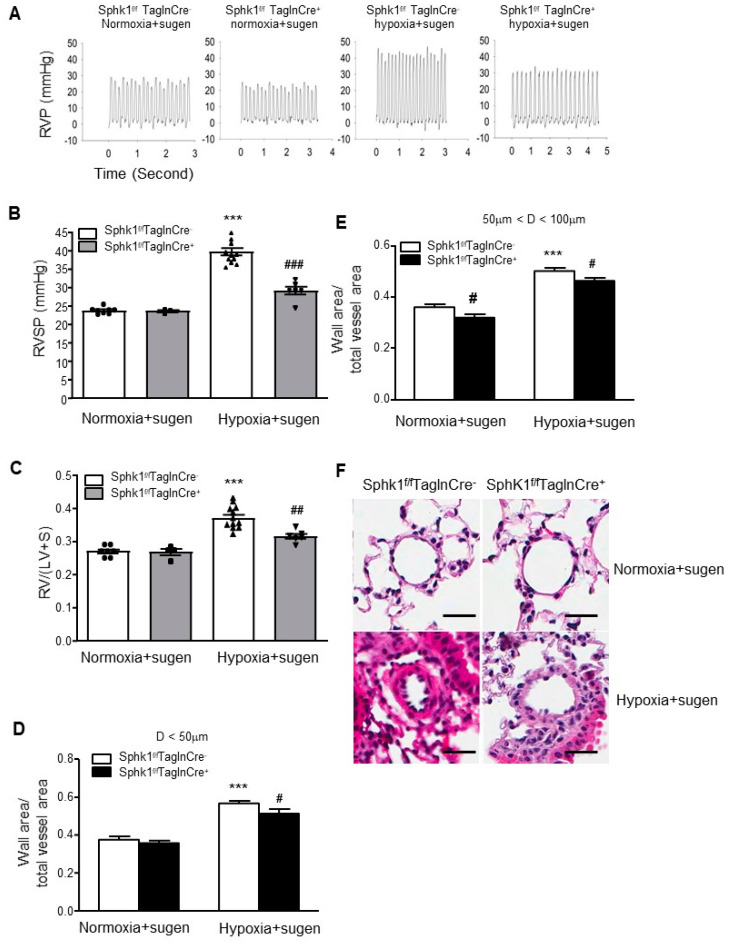
Smooth muscle cell-specific *Sphk1* knockout (*Sphk1^f/f^TaglnCre^+^*) mice are protected from hypoxia-plus-Sugen-induced pulmonary hypertension. Four doses of Sugen (20 mg/kg body weight) were given subcutaneously to the mice once per week. When compared with their wild-type siblings (*Sphk1^f/f^TaglnCre^−^*), *Sphk1^f/f^TaglnCre^+^* mice exposed to hypoxia plus Sugen developed (**A**,**B**) less severe elevations in right ventricular systolic pressure (RVSP), (**C**) less right ventricular hypertrophy (RV/(LV + S)), and lower increases in the ratios of wall area to total vessel area of pulmonary arteries less than 50 μm (**D**) and 50–100 μm in diameter (**E**). (**F**) Representative pulmonary artery images in the lung sections of control and *Sphk1^f/f^TaglnCre^+^* mice exposed to normoxia or hypoxia. Scalebar: 20 μm. Results are expressed as mean ± SEM; number of mice in each group is shown (n = 4–10) in (**B**,**C**). *** *p* < 0.001 versus normoxia *Sphk1^f/f^TaglnCre^−^* group; #, ### *p* < 0.05, ## *p* < 0.01, hypoxia *Sphk1^f/f^TaglnCre^−^* vs. hypoxia *Sphk1^f/f^TaglnCre^+^* groups.

**Figure 4 ijms-23-14516-f004:**
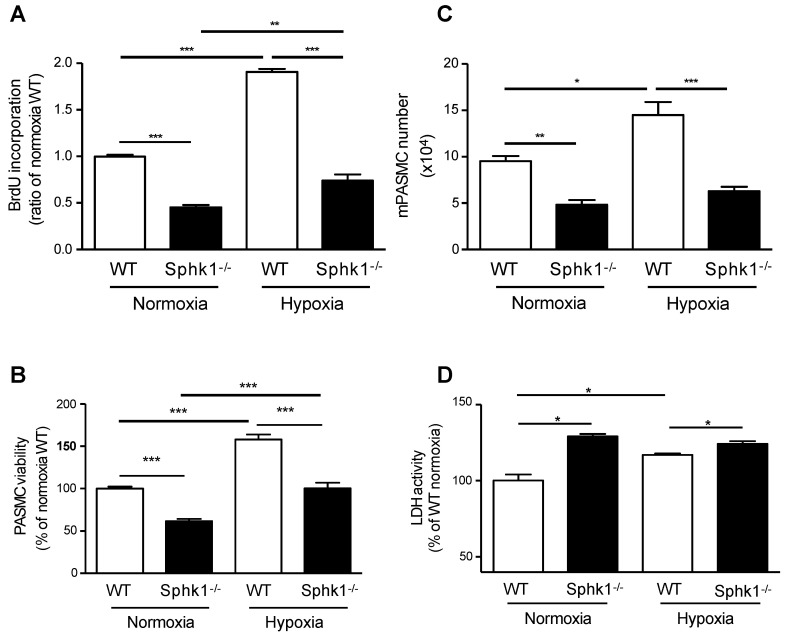
PASMCs isolated from *Sphk1* knockout (*Sphk1^-/-^*) mice are less proliferative than in WT mice under normoxia or hypoxia exposure. Cell proliferation was measured with a 5-bromo-2′-deoxyuridine (BrdU) incorporation assay (**A**), a viability assay (**B**), cell counting (**C**), and an LDH assay (**D**). Data are expressed as mean ± SEM *(n* = 4). * *p* < 0.05; ** *p* < 0.01; *** *p* < 0.001; bars indicate the comparison between groups.

**Figure 5 ijms-23-14516-f005:**
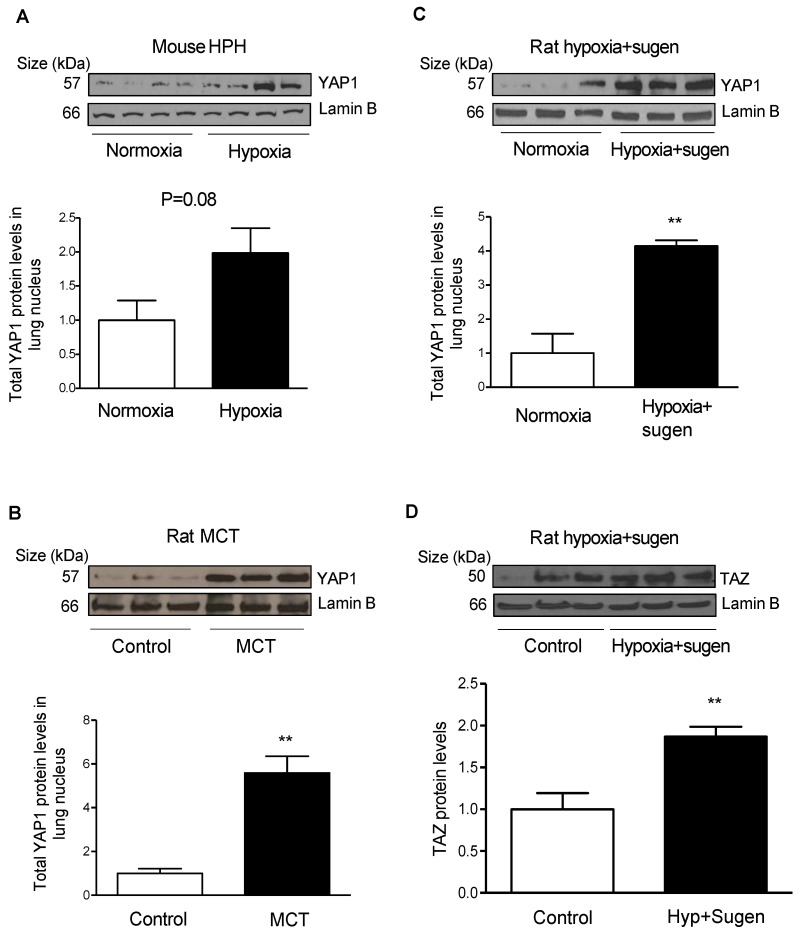
Upregulation of YAP1 and TAZ in rodent models of pulmonary hypertension. (**A**) Representative Western blot images and Lamin B-normalized quantification of protein demonstrate that YAP1 expression has an increasing trend in mouse lungs after 4 weeks of hypoxia exposure; (**B**) representative Western blot images and Lamin B-normalized quantification of protein demonstrate that YAP1 expression increased significantly in lung tissues from monocrotaline (MCT)-mediated pulmonary hypertension in rats; (**C**) representative Western blot images and Lamin B-normalized quantification of protein demonstrate that YAP1 expression increased significantly in lung tissues from hypoxia + Sugen-mediated pulmonary hypertension in rats; (**D**) representative Western blot images and Lamin B-normalized quantification of protein demonstrate that TAZ expression increased significantly in lung tissues from hypoxia + Sugen-mediated pulmonary hypertension in rats. ** *p* < 0.01 versus controls, *n* = 3–4 per group.

**Figure 6 ijms-23-14516-f006:**
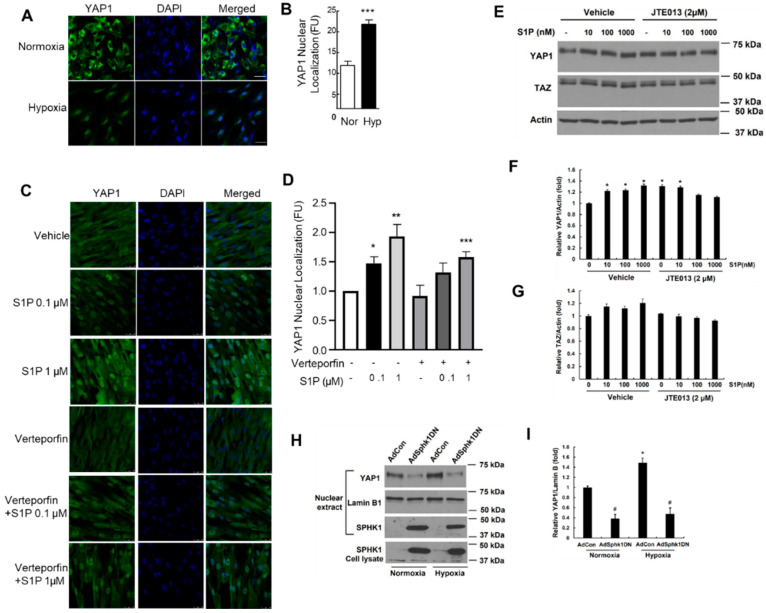
Hypoxia and S1P stimulate YAP1 nuclear translocation and verteporfin attenuates S1P-mediated YAP1 nuclear translocation in PASMCs. Exposing hPASMCs to hypoxia (3%, 30 min) significantly increased YAP1 translocation to the nucleus as determined by immunofluorescence (**A**), and nuclear YAP1 expression levels were quantified and are shown in (**B**). Similarly, the S1P (0.1 µM and 1 µM) treatment of PASMCs for 30 min induced YAP1 nuclear translocation, an effect that was abrogated by the pretreatment of cells with verteporfin (100 nM for 1 h), followed by an S1P challenge for 30 min (**C**,**D**). hPASMCs were pretreated with the S1PR2 antagonist JTE-013 (2 µM) for 1 h followed by S1P (10-1000 nM) challenge for 24 h. YAP1 and TAZ expressions were determined by Western blotting, and a representative blot is shown (**E**) and quantified by image analysis, and it is normalized to the total actin for YAP1 (**F**) and TAZ (**G**). hPASMCs were transfected with adeno control vector or SPHK1 dominant mutant (5 MOI, 24 h) prior to exposure to normoxia or hypoxia (3%, 2 h). Cells were subjected to the isolation of their nuclei using a commercial kit; subjected to Western blotting; and probed with anti-YAP1, anti-Lamin B, and anti-SPHK1 antibodies. The baseline and hypoxia-induced nuclear localization of YAP1 were reduced in hPASMCs transfected with SPHK1DN adenoviral plasmid compared with cells transfected with a control plasmid (**H**,**I**). *n* = 3–5. * *p* < 0.05 versus normoxia or vehicle control; ** *p* < 0.01 versus vehicle control; *** *p* < 0.05 versus S1P; # *p* < 0.05 versus vector control. Scale bar: 25 μm.

**Figure 7 ijms-23-14516-f007:**
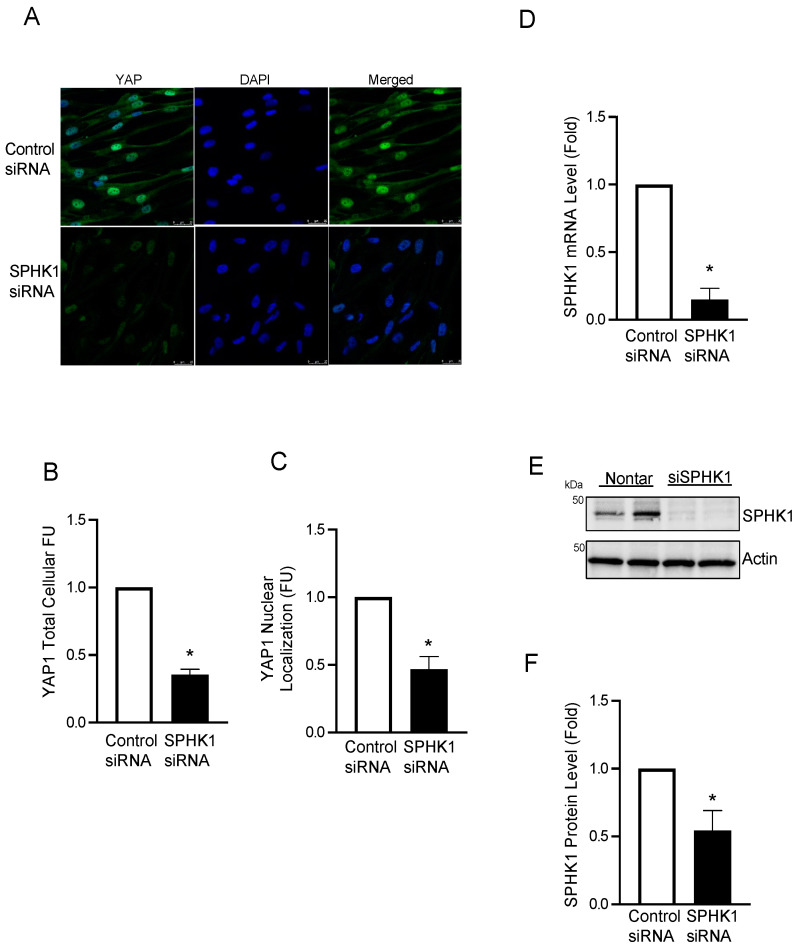
Silencing *SPHK1* via siRNA attenuates YAP1 expression and nuclear translocation. hPASMCs were treated with control or *SPHK1* siRNA (30 nM) for 72 h. (**A**–**C**) Immunofluorescence staining data and quantification data showing decreased total cellular and nuclear YAP1 levels; (**D**) real-time PCR showing decreased SPHK1 gene expression levels. (**E**,**F**) Representative Western blot images and β-actin-normalized quantification of protein demonstrate that SPHK1 protein levels were significantly silenced by siRNA. *n* = 3. * *p* < 0.05 versus control siRNA. Scale bar: 25 μm.

**Figure 8 ijms-23-14516-f008:**
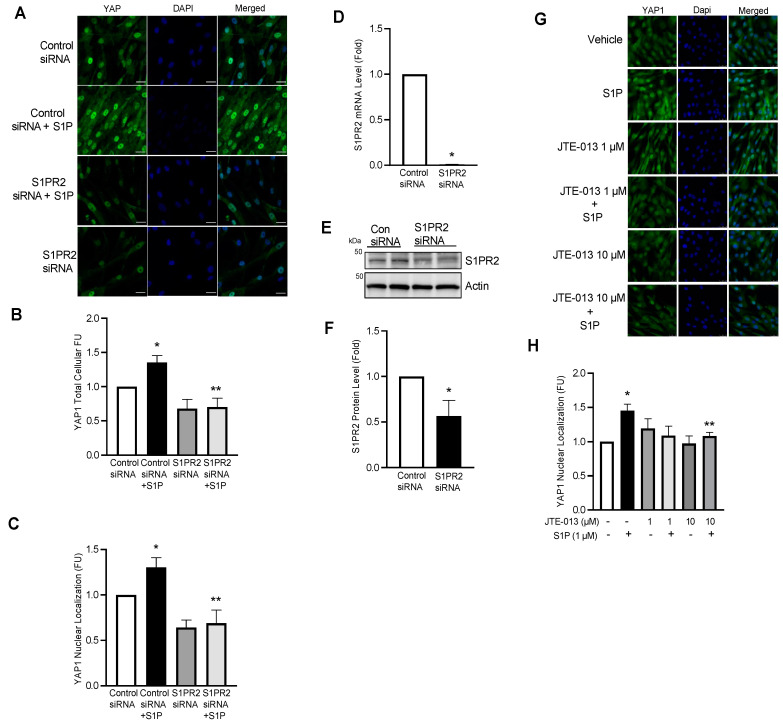
S1P promotes YAP1 nuclear translocation via ligation to S1PR2. hPASMCs were treated with control or S1PR2 siRNA (30 nM) for 72 h and then stimulated with S1P (1 µM, 24 h). (**A**–**C**) Immunofluorescence staining of total cellular and nuclear YAP1 in hPASMCs and quantification data demonstrating that SIPR2 knockdown inhibits S1P-induced YAP1 expression and nuclear translocation; (**D**) real-time PCR showing decreased S1PR2 gene expression; (**E**,**F**) representative Western blot images and β-actin-normalized quantification of protein demonstrate that S1PR2 protein levels were significantly silenced by siRNA; (**G**,**H**) immunofluorescence staining and quantification data showing that the S1PR2 inhibitor JTE-013 (10 µM, 1 h pretreatment) inhibits the S1P (1 µM, 24 h)-induced nuclear localization of YAP1. *n* = 3; * *p* < 0.05 versus siRNA or vehicle control; ** *p* < 0.05 versus siRNA control or versus S1P alone. Scalebar: 25 μm.

**Figure 9 ijms-23-14516-f009:**
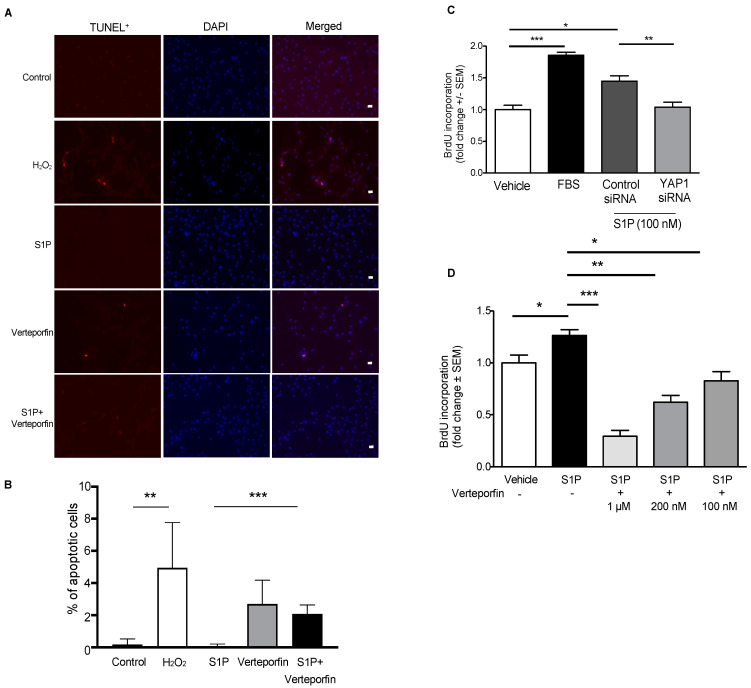
YAP1 regulates S1P-mediated proliferation in hPASMCs. (**A**) Representative images and (**B**) quantitative data from TUNEL assays. hPASMCs were cultured on a coverslip in a six-well cell culture plate. H_2_O_2_ (100 µM) was used as an apoptosis stimulus for hPASMCs. When cell confluence reached ~95%, cells were starved for 3 h, pre-treated with verteporfin (200 nM, 1 h)) and challenged with H_2_O_2_ or S1P (1 µM) for 24 h. Cell apoptosis was examined using an *in situ* BrdU–Red DNA fragmentation (TUNEL) kit. Representative images from each group are demonstrated. Scalebar: 20 µm. ** *p* < 0.01; *** *p* < 0.001. hPASMC cell proliferation was examined with BrdU assays; FBS (10%) was used as a positive control. (**C**) Silencing YAP1 with YAP1 siRNA (50 nM, 72 h) attenuated S1P-mediated hPASMC proliferation. (**D**) YAP1 inhibition with verteporfin (200 nM) attenuated S1P-mediated hPASMC proliferation in a dose-dependent manner. * *p* < 0.05; ** *p* < 0.01; *** *p* < 0.001. *n* = 3. Scale bar: 50 μm.

**Figure 10 ijms-23-14516-f010:**
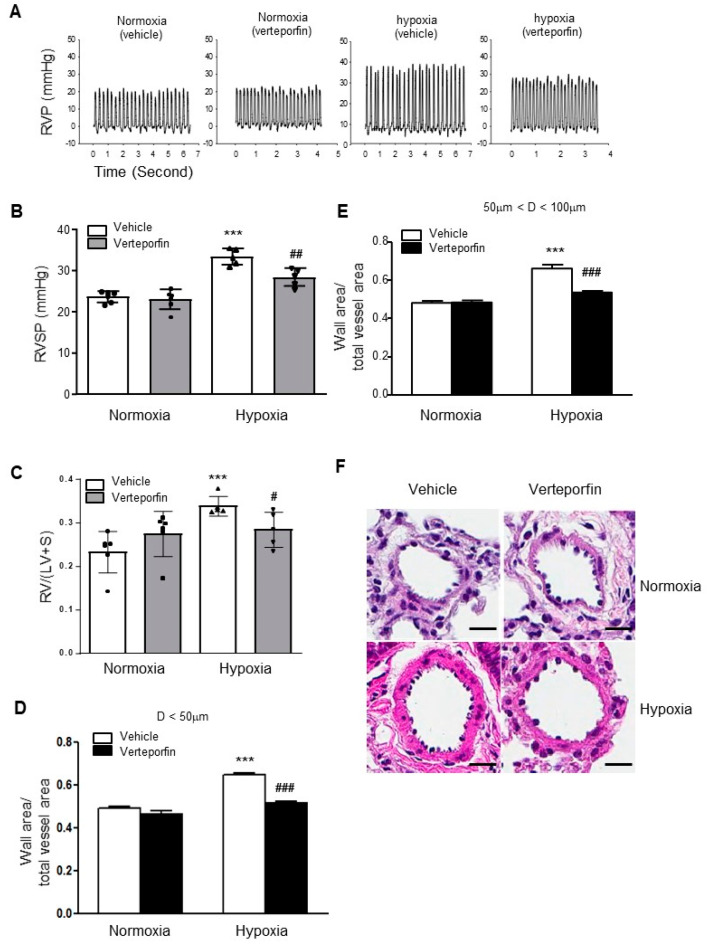
YAP1 inhibition with verteporfin prevents hypoxia-mediated PH and pulmonary vascular remodeling in mice. (**A**) Representative record of right ventricular pressure (RVP); (**B**) changes in right ventricular systolic pressure (RVSP); (**C**) changes in right ventricular hypertrophy (RV/(LV + S); (**D**) changes in ratios of wall area to total vessel area of pulmonary arteries less than 50 μm in diameter in the lung sections of control and verteporfin-treated groups after normoxia or hypoxia exposure; (**E**) changes in ratios of wall area to total vessel area of pulmonary arteries less than 50–100 μm in diameter in the lung sections of control and verteporfin-treated groups after normoxia or hypoxia exposure; (**F**) representative pulmonary artery images in the lung sections of control and verteporfin-treated groups after normoxia or hypoxia exposure. Bar size: 10 μm. Results are expressed as mean ± SEM; number of mice in each group (*n* = 5–6) is shown in (**B**,**C**). *** *p* < 0.001 versus normoxia without verteporfin group; # *p* < 0.05; ## *p* < 0.01; ### *p* < 0.001 versus hypoxia without verteporfin group.

**Figure 11 ijms-23-14516-f011:**
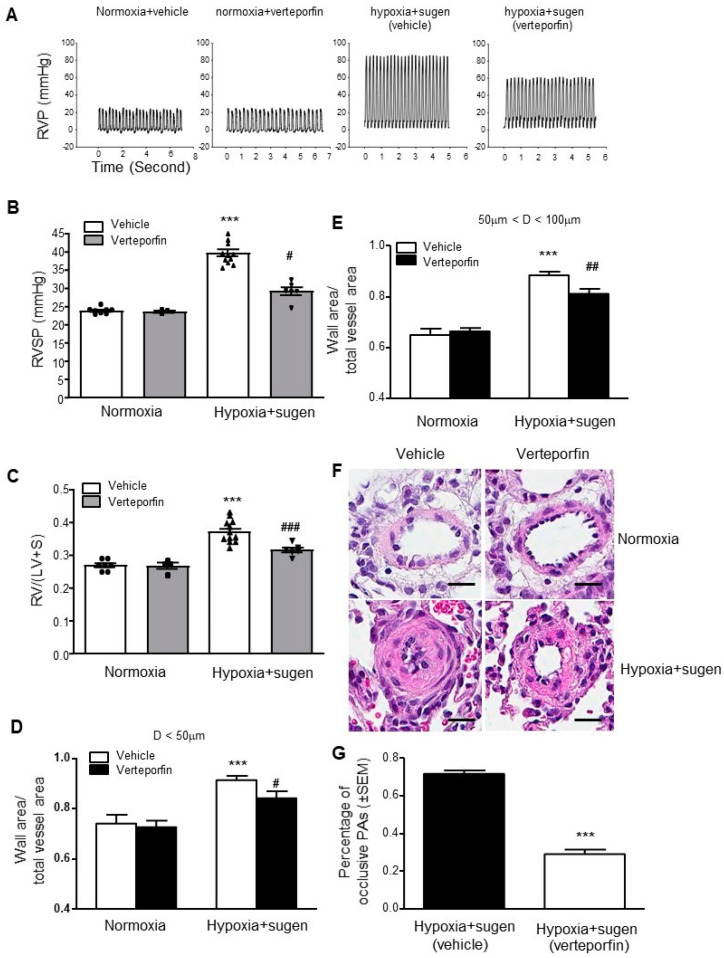
YAP inhibition with verteporfin attenuates hypoxia-plus-Sugen-mediated pulmonary hypertension in rats. Compared with the controls, verteporfin treatment attenuated the development of pulmonary artery hypertension (PAH), as assessed by changes in right ventricular systolic pressure (RVSP), right ventricular hypertrophy (RV/LV+S), and the development of medial hypertrophy and neointima formation. (**A**) Representative record of right ventricular pressure (RVP) tracings; (**B**) changes in RVSP; (**C**) changes in RV/(LV + S); (**D**) changes in ratios of wall area to total vessel area of pulmonary arteries less than 50 μm in diameter in the lung sections of the control and verteporfin-treated groups; (**E**) changes in ratios of wall area to total vessel area of pulmonary arteries 50–100 μm in diameter in the lung sections of the control and verteporfin-treated groups; (**F**) representative pulmonary artery images in the lung sections of control and verteporfin treated groups; (**G**) percentage of occlusive PAs significantly decreases with verteporfin therapy. Scale bar: 20 μm. Results are expressed as mean ± SEM; number of rats in each group (*n* = 4–10) is shown in panels B and C. *** *p* < 0.001 versus normoxia without verteporfin group (**B**–**E**) or hypoxia without verteporfin group (**G**); # *p* < 0.05; ## *p* < 0.01; ### *p* < 0.001 versus hypoxia without verteporfin group.

## Data Availability

Data available upon requests.

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
