# Peer review of "Sphingosine Kinase 1 Deficiency in Smooth Muscle Cells Protects against Hypoxia-Mediated Pulmonary Hypertension via YAP1 Signaling"

_ijms, 2022, doi:10.3390/ijms232314516_

Round 1
Reviewer 1 Report (Previous Reviewer 1)
The authors improved their manuscript substantially.
I have no further requests.
Author Response
Thank you.
Reviewer 2 Report (New Reviewer)
The authors investigated the pathogenetic role of sphingosine kinase 1 (SPHK1) found in pulmonary artery smooth muscle cells (PASMCs) in the development of pulmonary artery hypertension (PAH). They concluded that the activation of yes associated protein 1 (YAP1) signaling mediated via the SPHK1/sphingosine-1-phosphatase (S1P)/S1P receptor-2 (S1PR2) pathway in PASMCs is involved in pulmonary artery remodeling in PAH. They have also suggested the potential application of these signaling molecules as therapeutic targets for PAH. It is an interesting study with novel findings that can help expand our understanding of this field. However, I have some comments and suggestions for further improvement.
Major comments:
1. Exposure to hypoxia and treatment of human PASMCs with S1P induced the nuclear translocation of YAP1. Was a similar phenomenon observed in the PASMCs isolated from wild-type mice? What of sphk1f/f TaglenCre+ mice?
If the authors’ hypothesis that SPHK1/S1P/YAP1 signaling is involved in the pathogenesis of the PAH is correct, the conditional knockout of SPHK1 in smooth muscles may attenuate the hypoxia- and S1P-induced YAP1 nuclear translocation in PASMCs. I believe data supporting these points is essential to corroborate the authors’ conclusion.
2. The authors showed that hypoxia stimulated the expression of YAP1 and TAZ, while S1P enhanced YAP1 but not TAZ expression in human PASMCs. The results may imply that an S1P-independent pathway is involved in the hypoxia-induced upregulation of TAZ. Please consider discussing the possible mechanisms underlying this phenomenon.
3. The description of the statistical analysis used in this study is poor. There is no clarity on the post-hoc testing following ANOVA. Please clear up which statistical analysis was used for each data type; I believe, some of the data must have been analyzed with two-way ANOVA.
4. There was no description of the anesthetic procedure used during hemodynamic measurements. A procedure for sacrificing the animals was also not appropriately described. Please add proper detailed descriptions for all aforementioned aspects in the Methods section 4.7.
5. The description of the experimental procedure is too detailed to be given in a figure caption (for instance, in the figure caption given in lines 458–467). Please only retain the most relevant details and make it concise.
6. In Supplemental Figure 1C, please consider using ‘arrows’ instead of ‘arrowheads’. Also, please correct the typo “arrows. enote” to “arrows denote” in the figure caption.
7. “sphk1-/-” should be “sphk1f/f TaglenCre+” in the Result section 2.4. Did the author use global knockout of sphk1 in this section? Please clarify.
Minor comments:
1. Please make sure all abbreviations are defined upon first use. Some instances require attention; for example, ‘PH’ in line 25, ‘HPH’ in line 82, and ‘MCT’ in line 120.
2. The use of ‘and’ or ‘/’ between ‘hypoxia’ and ‘sugen’ in line 94 seems missing. Please provide a citation to the source about the hypoxia/sugen-induced PH model mice for the readers’ reference.
3. Please provide a citation for the nuclear translocation of YAP1 described in lines 127–128.
4. Some of the letters in the figures are not adequately visible; for example, ‘N’ of normoxia in Figure A and ‘B’ in Figure 6B.
5. The ‘2’ in ‘O2’ in line 329 must be in subscript.
Author Response
We thank the reviewer for the suggestions. Our responses are detailed below. Edited and new text are marked in red.
The authors investigated the pathogenetic role of sphingosine kinase 1 (SPHK1) found in pulmonary artery smooth muscle cells (PASMCs) in the development of pulmonary artery hypertension (PAH). They concluded that the activation of yes associated protein 1 (YAP1) signaling mediated via the SPHK1/sphingosine-1-phosphatase (S1P)/S1P receptor-2 (S1PR2) pathway in PASMCs is involved in pulmonary artery remodeling in PAH. They have also suggested the potential application of these signaling molecules as therapeutic targets for PAH. It is an interesting study with novel findings that can help expand our understanding of this field. However, I have some comments and suggestions for further improvement.
Major comments:
- Exposure to hypoxia and treatment of human PASMCs with S1P induced the nuclear translocation of YAP1. Was a similar phenomenon observed in the PASMCs isolated from wild-type mice? What of sphk1f/f TaglenCre+ mice? If the authors’ hypothesis that SPHK1/S1P/YAP1 signaling is involved in the pathogenesis of the PAH is correct, the conditional knockout of SPHK1 in smooth muscles may attenuate the hypoxia- and S1P-induced YAP1 nuclear translocation in PASMCs. I believe data supporting these points is essential to corroborate the authors’ conclusion.
We agree with the reviewer that performing these studies in Sphk1 deficient PASMCs would add incremental confirmation to our findings showing that YAP1 nuclear translocation is involved in SPHK1/S1P induced experimental PH. I do, however, disagree that these data are essential. In future studies we plan to use PASMCs from this model to further define the signal pathways and molecular mechanisms by which SPHK1/S1P/YAP1 promotes vascular remodeling and experimental PH. In this manuscript, we have used multiple in vivo and in vitro models which, together, demonstrate that YAP1 nuclear localization is involved in SPHK1/S1P induced vascular remodeling and PH. Using 2 experimental animal models of PH we show that Sphk1 deficiency in smooth muscle cells (Fig 1-3) as well as YAP1 inhibition (Fig 10-11) are protective against PH development in vivo. We further demonstrate that YAP1 and Taz expression in the lung are increased (Fig 5) in these PH models. Our ex vivo (Fig 4) and in vitro (Fig 6-9) studies performed in PASMCs demonstrate that the SPHK1/S1P/S1PR2 signaling axis and hypoxia leads to YAP1 nuclear localization and PASMC proliferation and that inhibition of YAP1 mitigates S1P induced PASMC proliferation.
- The authors showed that hypoxia stimulated the expression of YAP1 and TAZ, while S1P enhanced YAP1 but not TAZ expression in human PASMCs. The results may imply that an S1P-independent pathway is involved in the hypoxia-induced upregulation of TAZ. Please consider discussing the possible mechanisms underlying this phenomenon.
We agree that there may be some signal specific differences in hypoxia and S1P regulation of the YAP1/TAZ. This has been added to the discussion. Lines 236-241.
- The description of the statistical analysis used in this study is poor. There is no clarity on the post-hoc testing following ANOVA. Please clear up which statistical analysis was used for each data type; I believe, some of the data must have been analyzed with two-way ANOVA.
We thank the review for the suggestion. We used Dunnett’s test for multiple comparisons. The text was adjusted accordingly.
We performed one-way ANOVA as we were evaluating differences between groups considering essentially one factor as the independent factor and not the interaction between two factors. For example, in the in vivo hypoxic studies we were interested in testing differences between different genotypes and not the interaction effect of hypoxia and normoxia between different genotypes. Finally, it is important to point out that our analytical approach is very commonly used in similar studies published in the literature.
- There was no description of the anesthetic procedure used during hemodynamic measurements. A procedure for sacrificing the animals was also not appropriately described. Please add proper detailed descriptions for all aforementioned aspects in the Methods section 4.7.
We thank the reviewer for the suggestion. The text was modified accordingly.
- The description of the experimental procedure is too detailed to be given in a figure caption (for instance, in the figure caption given in lines 458–467). Please only retain the most relevant details and make it concise.
The figure legends have been modified to remove procedures that are repeated in the Materials and Methods section.
- In Supplemental Figure 1C, please consider using ‘arrows’ instead of ‘arrowheads’. Also, please correct the typo “arrows. enote” to “arrows denote” in the figure caption.
Thank you for correcting the typo. Arrows have been added to the figure.
- “sphk1-/-” should be “sphk1f/f TaglenCre+” in the Result section 2.4. Did the author use global knockout of sphk1in this section? Please clarify.
Global Sphk1 knockout mice were used for these experiments. This has been clarified in line 108.
Minor comments:
- Please make sure all abbreviations are defined upon first use. Some instances require attention; for example, ‘PH’ in line 25, ‘HPH’ in line 82, and ‘MCT’ in line 120.
We have defined these at first use on lines 25 and 72.
- The use of ‘and’ or ‘/’ between ‘hypoxia’ and ‘sugen’ in line 94 seems missing. Please provide a citation to the source about the hypoxia/sugen-induced PH model mice for the readers’ reference.
Thank you, the correction has been made. Both the original paper and a paper demonstrating our use of the hypoxia/sugen model have been referenced in the methods section 4.7.
Ciuclan L, Bonneau O, Hussey M, et al. A novel murine model of severe pulmonary arterial hypertension. Am J Respir Crit Care Med 2011;184:1171–1182.
Chen J, Sysol JR, Singla S, Zhao SP, Yamarua A, Valdez-Jasso D, Abbasi T, Shioura KM, Sahni S, Reddy V, Sridhar A, Camp SM, Tang H, Gao H, Ye SQ, Comhair S, Dweik R, Hassoun P, Yuan J X-J, Garcia JGN, Machado RF. Nicotinamide Phosphoribosyltransferase Promotes Pulmonary Vascular Remodeling and is a Therapeutic Target for Pulmonary Arterial Hypertension. Circulation. 2017; 135(16):1532-1546.
- Please provide a citation for the nuclear translocation of YAP1 described in lines 127–128.
The following reference has been included:
Basu S, Totty NF, Irwin MS, Sudol M, Downward J. Akt Phosphorylates the yes-associated protein, YAP, to induce interaction with 14-3-3 and attenuation of p73-mediated apoptosis. Mol Cell. 2003;11(1):11. DOI:10.1016/S1097-2765(02)00776-1.
- Some of the letters in the figures are not adequately visible; for example, ‘N’ of normoxia in Figure A and ‘B’ in Figure 6B.
The figure has been corrected.
- The ‘2’ in ‘O2’ in line 329 must be in subscript.
Thank you we have made the correction.
Round 2
Reviewer 2 Report (New Reviewer)
Thank you for your reply and for addressing my comments. The manuscript has been improved, but the authors should revise the manuscript by addressing the following comments.
Major comments:
1.
>>The description of the statistical analysis used in this study is poor. There is no clarity on the post-hoc testing following ANOVA. Please clear up which statistical analysis was used for each data type; I believe, some of the data must have been analyzed with two-way ANOVA.
>We thank the review for the suggestion. We used Dunnett’s test for multiple comparisons. The text was adjusted accordingly.
Unfortunately, I could not find the revised text regarding these points. Please clearly describe them in the Method section 4.8. or somewhere else.
2.
>>There was no description of the anesthetic procedure used during hemodynamic measurements. A procedure for sacrificing the animals was also not appropriately described. Please add proper detailed descriptions for all aforementioned aspects in the Methods section 4.7.
>We thank the reviewer for the suggestion. The text was modified accordingly.
I could not find the modified texts in the revised manuscript. Please clearly declare the anesthetic procedures for the mice and rats during hemodynamic measurements and the procedures for sacrificing the animals in the text.
Minor comments:
1.
>>In Supplemental Figure 1C, please consider using ‘arrows’ instead of ‘arrowheads’. Also, please correct the typo “arrows. enote” to “arrows denote” in the figure caption.
>Thank you for correcting the typo. Arrows have been added to the figure.
Please delete “.” from “Red arrows.” in line 842.
2.
>>Some of the letters in the figures are not adequately visible; for example, ‘N’ of normoxia in Figure A and ‘B’ in Figure 6B.
>The figure has been corrected.
Please check Figure 1A.
3.
>>The use of ‘and’ or ‘/’ between ‘hypoxia’ and ‘sugen’ in line 94 seems missing. Please provide a citation to the source about the hypoxia/sugen-induced PH model mice for the readers’ reference.
>Thank you, the correction has been made. Both the original paper and a paper demonstrating our use of the hypoxia/sugen model have been referenced in the methods section 4.7.
>Ciuclan L, Bonneau O, Hussey M, et al. A novel murine model of severe pulmonary arterial hypertension. Am J Respir Crit Care Med 2011;184:1171–1182.
This paper is not included in the reference list.
Author Response
Our sincere apologies to the reviewer.
It appears that the wrong version of the manuscript was uploaded to the site.
This has been corrected.

Round 3
Reviewer 2 Report (New Reviewer)
Please make sure that the authors properly revised the manuscript by addressing all of the comments from the reviewer before submission.
Minor comments:
1.
>>>In Supplemental Figure 1C, please consider using ‘arrows’ instead of ‘arrowheads’. Also, please correct the typo “arrows. enote” to “arrows denote” in the figure caption.
>>Thank you for correcting the typo. Arrows have been added to the figure.
>Please delete “.” from “Red arrows.” in line 842 (line 849 of the latest version).
It is not corrected.
2.
>>>Some of the letters in the figures are not adequately visible; for example, ‘N’ of normoxia in Figure A and ‘B’ in Figure 6B.
>>The figure has been corrected.
>Please check Figure 1A.
It is not corrected.
Author Response
This has been corrected

This manuscript is a resubmission of an earlier submission. The following is a list of the peer review reports and author responses from that submission.
Round 1
Reviewer 1 Report
This study describes a protective role of sphingosine kinase 1 deficiency in smooth muscle cells against hypoxia-mediated pulmonary hypertension. The study is based on previous observations of an increased expression of SphK1 in PASMCs isolated from patients with PAH. The genetic deletion of SphK1 or pharmacological inhibition of its activity was shown to diminish hypoxia-induced PH in rodents.
In the present study the authors elegantly demonstrate a pivotal role of YAP1 signaling in the context of SPHK1/S1PR2/S1P axis in PASMCs and propose YAP1 inhibition as a novel target for PH therapy.
The study is well designed, well performed and comprehensively described.
I have only minor comments:
- Did the authors measure the expression or activity of SphK2, Spp1/2 or Sgpl in SphK1 KO cells under conditions chosen for this study?
- Please specify the number of individual experiments performed in each Figure legend.
- In the Discussion specify “YAP” or “YAP1”? (p 16, l 314-328)
Reviewer 2 Report
The authors present data that demonstrate slight protection from pulmonary hypertension with the loss of Sphk1 in smooth muscle cells.
In the results the data need to describe each panel in each figure, not just conclude the findings. The results should state the premise for the experiment, describe the data and conclude with a sentence that 99% of the readers would agree with. The description of Figure 3 does this much better than 1 or 2 but still needs improvement.
Furthermore, the authors cannot make this statement without directly comparing the differences in the models used for Figures 2 and 3. "Compared to hypoxia-mediated PH model in Figure 2, control mice exhibited significantly higher RVSP, and RVH values and more PVR in the hypoxia-Sugen mediated PH model." Especially given that the RVSP and RVH values are very similar between figures.
No Figure Legend for Figure S1. This is critical in order to understand what the arrows indicate in panel C.
Data in Figure 4 demonstrate that SphK1 regulates growth, viability and LDH release under normoxia, suggesting a role for SphK1 basally in these processes. However, the overall change from normoxic to hypoxic conditions seems to be of the same magnitude for WT and SphK1-/- cells. The authors should examine and acknowledge the basal processes. Moreover, the last sentence describing Figure 4 states "serum-induced" processes, but this is not described clearly in the text.
This statement is misleading, as panel A is not significant. "As shown in Figure 5 A-C, lung tissues from mouse HPH, rat MCT and rat Sugen + Hypoxia showed significantly higher levels of YAP1 expression compared to control lungs."
Figures 6 and 8. The authors use 1uM S1P, which is unfortunately used in the literature; however, is orders of magnitude over the Kd for the receptor, the authors should include a dose response and describe this limitation in the discussion. This is especially important as they use 100nM in Figure 9, and show greater response than with 1uM.
Figure 6C. YAP1 appears more perinuclear than nuclear, please clarify/explain, perhaps with higher magnification images.
Figures 7. YAP1 seems to have lower overall expression in cells treated with SphK1 siRNA, this should be probed via western. With that said, the western blot in Panel 7D is not acceptable.
Figure 8 Panel A, DAPI is very faint in the control siRNA+S1P and colocalization is very difficult to discern. Based on the images shown, YAP1 is nuclear in the S1PR2 siRNA panels. The S1PR2 antagonist JTE-013 should also be used to demonstrate the necessity of S1PR2, and S1PR1 and 3 antagonist VPC2301 should b used to rule out the involvement of other S1PRs.
Reviewer 3 Report
This study reports on the involvement of the SPHK1/S1P signaling axis in hypoxia-induced YAP1 and Taz expression, their contribution to smooth muscle cell proliferation and survival, and ultimately to the disorder pulmonary hypertension. They have performed rat and mouse models of hypoxia-induced PH and show that SPHK1 is upregulated in smooth muscle cells in PH, while SMA-specific Sphk1 knockout mice were protected from hypoxia-induced PH. The same was seen in a model of hypoxia+sugen-induced PH. They further showed that isolated PASMCs from Sphk1 KO mice proliferated less and were less viable and showed more cytotoxicity. PH mice and rats showed also increased expression of YAP1 and Taz and in cell culture, this was mimicked by hypoxia exposure and by S1P stimulation via S1P2. Finally, they used a YAP1 inhibitor (verteporfin) that reduced S1P-stimulated proliferation, induce apoptosis in PASMCs, and also reduced vascular pathologies in the PH model.
Major points:
1.In Fig. 5D, they show that Taz is also increased in the rat PH model. Is this also seen in the mouse model and is it mediated by SPHK1/S1P? Show Taz in either Western blots or Immunofluorescence in cells stimulated with S1P.
2.In Fig. 8C, the downregulation of S1P2 by siRNA is not convincing. The calculated 50% downregulation is not seen in the Western blot and does not fit to the very clear reduction of Sphk1 by siRNA seen in the Western blot and also calculated as 50% in Fig. 7C. Also, be aware that S1P2 antibodies are not convincingly validated in Western blots. The authors should rather show the downregulation on mRNA level. In addition, the authors should use the selective S1P2 antagonist JTE-013 to strengthen their hypothesis that S1P2 is involved in the YAP1 upregulation.
3.In Fig. 9B and D, they show that the YAP1 inhibitor verteporfin increases apoptosis and reduces proliferation in the presence of S1P. In these experiments, the effect of verteporfin in the absence of S1P is missing. Verteporfin is known to be converted to cytotoxic metabolites, and it would therefore be interesting to see if S1P can reduce the verteporfin-induced apoptosis. For this, verteporfin alone is needed.
4.Since sugen is known as a VEGFR inhibitor, and in view of the fact that hypoxia can induce VEGF, then acting on VEGFR, it would be interesting to see if SPHK1 expression is altered in the setting: normoxia, hypoxia, hypoxia+sugen. Also explain, when first time mentioned, what sugen is and its primary mechanism of action.
5.Check and confirm that the single data points in Fig. 2C, columns 1 and 2 (representing normoxia) are different from the single data points in Fig. 3C, columns 1 and 2 (representing hypoxia + sugen). The look very similar.